# Biomarkers to predict or measure steroid resistance in idiopathic nephrotic syndrome: A systematic review

Carl J. May[1]*, Nathan P. Ford[2], Gavin I. Welsh[1], Moin A. Saleem[1,3]

1 Bristol Renal, University of Bristol, Bristol, United Kingdom, 2 University of Cape Town, Cape Town, South Africa, 3 Bristol Royal Hospital for Children, Bristol, United Kingdom

* carl.may@bristol.ac.uk

**Data Availability Statement:** All relevant data is contained within the paper.

**Funding:** The author(s) received no specific funding for this work.

## Abstract

In this systematic review we have sought to summarise the current knowledge concerning biomarkers that can distinguish between steroid-resistant nephrotic syndrome and steroid-sensitive nephrotic syndrome. Additionally, we aim to select biomarkers that have the best evidence-base and should be prioritised for further research. Pub med and web of science databases were searched using "steroid resistant nephrotic syndrome AND biomarker". Papers published between 01/01/2012 and 10/05/2022 were included. Papers that did not compare steroid resistant and steroid sensitive nephrotic syndrome, did not report sensitivity/specificity or area under curve and reviews/letters were excluded. The selected papers were then assessed for bias using the QUADAS-2 tool. The source of the biomarker, cut off, sensitivity/specificity, area under curve and sample size were all extracted. Quality assessment was performed using the BIOCROSS tool. 17 studies were included, comprising 15 case-control studies and 2 cross-sectional studies. Given the rarity of nephrotic syndrome and difficulty in recruiting large cohorts, case-control studies were accepted despite their limitations. We present a range of candidate biomarkers along with scores relating to the quality of the original publications and the risk of bias to inform future investigations. None of the selected papers stated whether the authors were blinded to the patient's disease when assessing the index test in the cohort. Highlighting a key problem in the field that needs to be addressed. These candidate biomarkers must now be tested with much larger sample sizes. Using new biobanks such as the one built by the NURTuRE-INS team will be very helpful in this regard.

## Introduction

The kidneys are responsible for many functions vital to sustaining life. They regulate blood pressure, monitor blood pH balance and remove waste products from the blood [1]. The glomerulus is the site of ultrafiltration where small solutes are excreted while proteins and macromolecules are retained [2]. This permselectivity is achieved thanks to the highly specialised structure of the glomerular filtration barrier [3]. The breakdown of the architecture of this

**Competing interests:** The authors have declared that no competing interests exist.

barrier leads to runaway proteinuria resulting in the clinical triad of oedema, hypoalbuminemia and proteinuria [4]. This collection of symptoms is termed nephrotic syndrome. It is often classified according to its apparent histopathological presentation. Nephrotic syndrome has many classifications. It can be primary, when the problem arises within the kidney itself, or secondary when the disease pathogenesis commences outside the kidney, as in lupus or HIV associated nephropathy. Primary nephrotic syndrome can be genetic or non-genetic. Nephrotic syndrome is the most common glomerular disease of childhood. It has an annual incidence between 1 and 17 cases per 100,000 [5–10]. There are currently over 70 genes that have been implicated in the pathogenesis of nephrotic syndrome. The pathogenesis of non-genetic or idiopathic nephrotic syndrome (INS) is not well understood. The seminal work of Shalhoub *et al* and many others since has demonstrated a role for a circulating permeability factor. This factor may be derived from either T-Cells [11, 12], B-Cells [13] or immature myeloid cells [14]. INS is treated with steroids. Steroid sensitive nephrotic syndrome (SSNS) has a very good prognosis with less than 5% progressing to chronic kidney disease [15]. However, between 10 and 20% of patients are steroid resistant (steroid resistant nephrotic syndrome, SRNS) and have a 50% risk of developing end-stage renal failure within 5 years of diagnosis [16]. Even amongst patients who do respond to steroid treatment a subset of these will progress to steroid-resistance end-stage renal failure patients requiring dialysis and or transplant. NS can also be characterised by its histopathological features. Focal Segmental Glomerulosclerosis (FSGS) progresses more rapidly to end-stage renal failure compared to minimal change disease (MCD) [17, 18]. Histopathological variants have limited correlation with the pathogenesis of the different NS entities, however, renal biopsies of SRNS generally show FSGS [19].

It is vital to preserve kidney function by using effective treatments as soon as possible. Currently steroid-resistant patients are identified by their lack of response to a course of steroid treatment. This exposes patients to the unnecessary side-effects of a futile treatment. There is a clear need to be able to differentiate between steroid-sensitive and steroid-resistant patients quickly and accurately.

The use of high-quality biomarkers that can distinguish between steroid-sensitive and steroid-resistant forms of idiopathic nephrotic syndrome would be a paradigm shift for nephrologists and their patients. Instead of being exposed to ultimately useless steroid treatment and enduring the side-effects, steroid-resistant patients could have a simple blood or urine test and proceed to treatment with secondary agents such as calcineurin inhibitors, alkylating agents, mycophenolate mofetil or rituximab.

To summarise what is currently known about potential biomarkers we carried out a systematic literature review. Then applied a quality appraisal tool to identify the most promising biomarker(s) for future more intensive efforts.

## Methods

### Eligibility criteria

Original research articles that compared biomarkers between known steroid sensitive and steroid resistant nephrotic syndrome patients were included. Review articles, conference proceedings, abstracts and letters to the editor were reviewed as a source for original studies but excluded from final review. Studies looking at individual candidate biomarkers and panels were included, but studies reliant on kidney biopsies were not included. A decision was made *a priori* to limit the review to biomarkers from blood, plasma, or urine samples, but not to focus on studies of kidney biopsies, which are invasive and pose a risk of harm for the patient. Studies were not excluded based on patient group characteristics beyond having a steroid sensitive and a steroid resistant group.

## Study screening and selection

Databases were searched for "Steroid Resistant Nephrotic Syndrome AND Biomarker". This strategy is deliberately narrow and prioritises specificity over selectivity. In this way we hoped to identify biomarkers that were directly relevant to steroid resistance. The search results from PubMed and Web of Science were exported to EndNote and screened for duplicates. Studies published between 1st January 2012 until 10th May 2022 were included to focus the review to recent techniques and results. Duplicates were omitted and the titles and abstract of the articles were screened, and the full text of potentially eligible studies was reviewed. Papers were screened by CM without using any automated tools.

## Data extraction

The sensitivity and specificity or Area Under the Curve (AUC) were extracted for each candidate biomarker or panel. The quality of the data was assessed using BIOCROSS [20] and bias and applicability was scored using QUADAS-2 [21]. CM collected the data and performed the bias and applicability assessment without using any automated tools.

The required characteristics were extracted and tabulated manually by CM and are shown in Table 1.

## Data synthesis and analysis

Data was synthesised and analysed using Excel (Microsoft) and Prism (Graphpad).

## BIOCROSS assessment

BIOCROSS is a quality assessment tool that is used to quantify the quality of the data that supports candidate biomarkers. The methodology of this tool is covered including details of the scoring is covered here [20]. Briefly, the tool includes 10-items covering 5 domains: 'Study rational', 'Design/Methods', 'Data analysis', 'Data interpretation' and 'Biomarker measurement', aiming to assess different quality features of biomarker cross-sectional studies. Each of the 10 items has three issues to consider. If each issue is covered then the publication will score 2 for that item, if only one or two of the issues are covered then the paper will score 1 and if none of the issues are covered then the score will be 0. A total score of 20 is available for papers that cover all the issues across all items and domains.

## QUADAS-2 assessment

The QUADAS-2 tool assesses the design and publication of biomarker data for applicability and risk of bias across four domains: patient selection, index test, reference standard, and flow and timing. It allows researchers to compare the risk of bias across different studies. The specific methodology is covered in detail here [21].

## Registration and protocol

This systematic review was not prospectively registered, and no review protocol has been made available.

## Missing data

The exclusion criteria were designed such that all included papers have the requisite data for inclusion in the synthesis.

**Table 1. Summary table of candidate biomarkers and panels.**

| Candidate | Biomarker | Paper | Source | Cut Off | Sensitivity | Specificity | AUC | Sample Size | BIOCROSS Score | Ref |
|---|---|---|---|---|---|---|---|---|---|---|
| 1 | Nephronectin | Watany et al, 2018 | Serum | 1.215 ng/ml | 74.2% | 92.0% | 0.896 | 40 SSNS 40 SRNS | 18/20 | [23] |
| 2 | A panel of Vitamin D Binding Protein, Adiponectin and MMP-2 | Agrawal et al 2020 | Plasma | NR | NR | NR | 0.78 | 24 SSNS 13 SRNS | 15/20 | [24] |
| 3 | A panel of Vitamin D Binding Protein, Neutrophil Gelatinase-Associated Lipocalin, Fetuin-1, Prealbumin, Alpha-1-Acid Glycoprotein 2, Acid Glycoprotein, Alpha-2-Macroglobulin, Alpha-1-B Glycoprotein, Thyroxine-Binding Globulin and Hemopexin | Bennet et al 2017 | Urine | N/A | 80.0 | 80.6 | 0.92 | 25 SSNS 25 SRNS | 14/20 | [25] |
| 4 | Vitamin D Binding Protein | Bennet et al 2016 | Urine | NR | 80.0 | 83.0 | 0.92 | 28 SSNS 24 SRNS | 14/20 | [26] |
| 5 | A panel of 50S Ribosomal Protein L32, S-Adenosylmethionine Decarboxylase α Chain, FK506-Binding Protein 1A and 30S ribosomal protein S11 | Bai et al 2013 | Urine | | 88.89% | 91.00% | NR | 32 SSNS 9 SRNS | 12/20 | [27] |
| 6 | Haptoglobin | Wen et al 2012 | Serum | 1.279 µg/ml | 85.0% | 96.3% | NR | 54 SSNS 52 SRNS | 16/20 | [28] |
| 7 | Vitamin D Binding Protein and Neutrophil Gelatinase-Associated Lipocalin | Choudhary et al 2020 | Urine | 303.81 ng/ml and 13.1 ng/ml | 82.0% and 86.0% | 78.0% and 89.0% | NR | 28 SSNS 28 SRNS | 16/20 | [29] |
| 8 | Urinary Protein Carbonyl Content | Gopal et al 2017 | Urine | 7.02 nmol/mg | 83.3% | 85.2% | 0.803 | SSNS 47 SRNS 23 | 14/20 | [30] |
| 9 | P-Glycoprotein and MRP-1 | Prasad et al 2021 | PBMCs | 7.13 and 9.62% | 90.0 and 80.7% | 80.0 and 90% | NR | SSNS 171 SRNS 83 | 17/20 | [31] |
| 10 | Soluble Urokinase Plasminogen Activator Receptor | Peng et al 2015 | Serum | NR | 73.5% | 79.2% | 0.80 | SSNS 108 SRNS 68 | 17/20 | [32] |
| 11 | Endothelin-1 | Ahmed et al 2019 | Serum | 24.6 pg/mL | 90.5% | 84.0% | 0.88 | 30 SSNS 25 SRNS | 14/20 | [33] |
| 12 | Soluble Urokinase Plasminogen Activator Receptor | Mousa et al 2018 | Serum | 33.17 ng/mL | 100% | 100% | 0.99 | 25 SSNS 25 SRNS | 14/20 | [34] |
| 13 | Urinary Protein Bound Sialic Acid | Gopal et al 2016 | Urine | 2.71 µg/mg | 75% | 75.5% | 0.814 | 47 SSNS 23 SRNS | 15/20 | [35] |
| 14 | Interleukin-7, Interleukin-9 and Monocyte Chemoattractant Protein-1 | Agrawal et al 2021 | Plasma | NR | 0.643 | 0.846 | NR | 26 SSNS 14 SRNS | 13/20 | [36] |
| 15 | Neutrophil Gelatinase-Associated Lipocalin /Creatinine ratio | Nickavar et al 2016 | Urine | 1.15ng/mg | 100% | 100% | NR | 25 SSNS 23 SRNS | 14/20 | [37] |
| 16 | N-Acetyl-beta-D Glucosaminodase /creatinine ratio | Mishra et al 2012 | Urine | 108.9u/g | 78.8% | 100.0% | NR | 10 FENS 8 SRNS | 15/20 | [38] |
| 17 | Interleukin-8 | Ahmed et al 2019 | Urine | 35.3 pg/mg | 93% | 85% | 0.94 | 40 SSNS 25 SRNS | 13/20 | [39] |

SRNS, steroid resistant nephrotic syndrome; SSNS, steroid sensitive nephrotic syndrome; BIOCROSS marks are out of 20 and scored across 5 domains: study rationale, design/methods, data analysis, data interpretation and biomarker measurement.

## Results

Adapted from Page et al [22].Most publications were excluded from the review because they either didn't compare steroid sensitive patients with steroid resistant patients, or they didn't report either sensitivity and specificity or AUC. After applying the inclusion/exclusion criteria and screening for eligibility, 17 studies were taken through to review, as hown in Fig 1. The most common sample was urine (9 studies) then serum (5 studies) then plasma (2 studies).

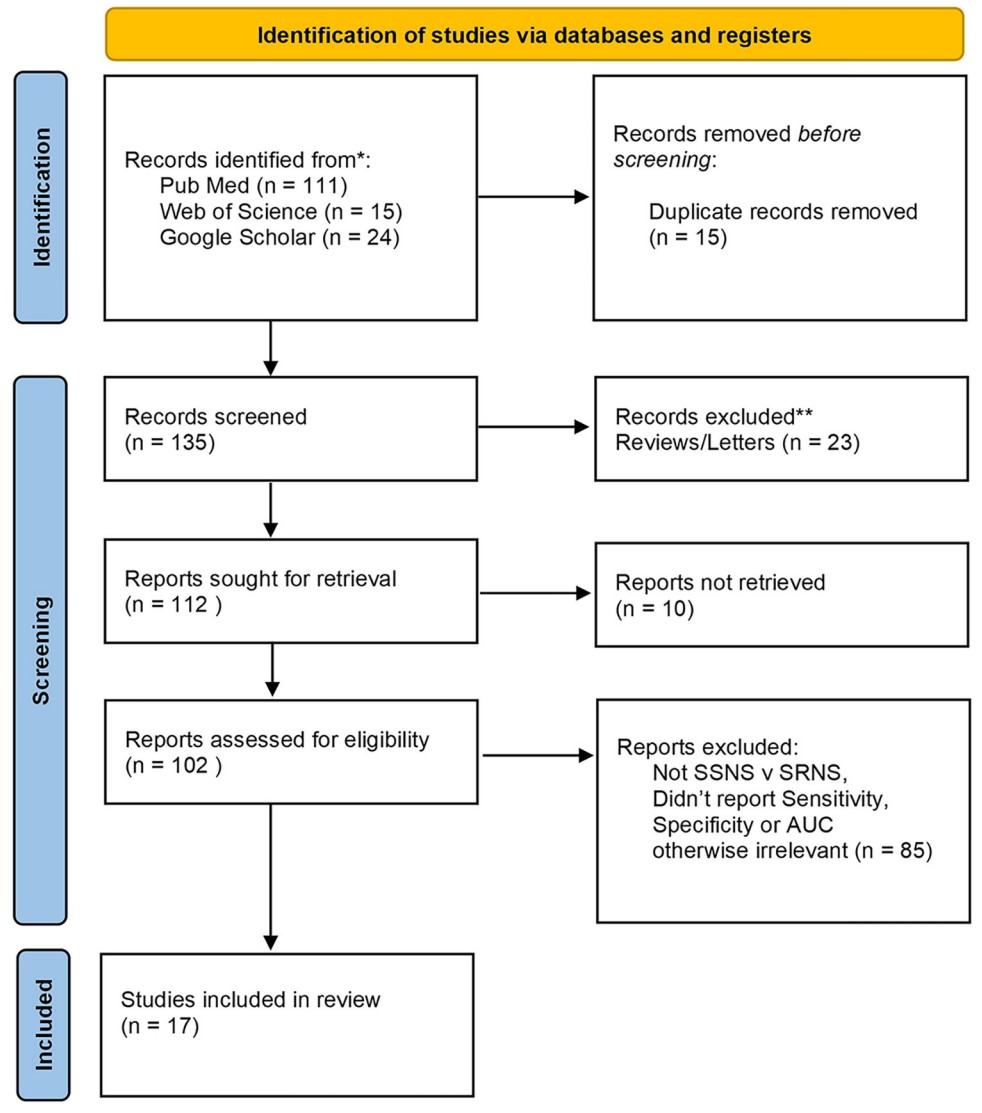

**Fig 1. PRISMA flow diagram.**

Table 1 shows the source paper and key descriptors for the candidate biomarker or panel. The BIOCROSS score indicates the quality of the source article, the higher the number the better the quality [20].

All of the identified studies fell afoul of the same reporting errors. None of them reported whether consecutive or random sampling was employed (Table 2). Similarly, there was not enough information provided by any of the identified manuscripts concerning blinding.

## Nephronectin

Nephronectin is a basal lamina protein found in glomerular basement membrane [40]. It is produced by the podocytes and is downregulated following podocyte injury [41]. In glomerular diseases such as focal segmental glomerulosclerosis and membranous nephropathy, nephronectin is also known to be downregulated [41]. Such are these changes in nephronectin expression during and following injury that nephronectin has been identified as a marker of

**Table 2. Responses to the QUADAS-2 assessment.**

| | Patient Selection | | | |
|---|---|---|---|---|
| | **Could the selection of patients have introduced bias?** | | | |
| Paper | SQ1 | SQ2 | SQ3 | Applicability |
| **1:** Watany *et al* | Not Reported | No | Yes | None |
| **2:** Agrawal *et al* | Not Reported | No | Not Reported | Unknown |
| **3:** Bennet *et al* | Not Reported | No | Yes | None |
| **4:** Bennet *et al* | Not Reported | No | Yes | None |
| **5:** Bai *et al* | Not Reported | No | Not Reported | Unknown |
| **6:** Wen *et al* | Not Reported | No | Yes | None |
| **7:** Choudhary *et al* | Not Reported | Yes | Yes | None |
| **8:** Gopal *et al* | Not Reported | No | Yes | None |
| **9:** Prasad *et al* | Not Reported | No | Yes | None |
| **10:** Peng *et al* | Not Reported | No | Yes | None |
| **11:** Ahmed *et al* | Not Reported | No | Yes | None |
| **12:** Mousa *et al* | Not Reported | No | Yes | None |
| **13:** Gopal *et al* | Not Reported | No | Yes | None |
| **14:** Agrawal *et al* | Not Reported | No | Not Reported | Unknown |
| **15:** Nickavar *et al* | Not Reported | Yes | Yes | None |
| **16:** Mishra *et al* | Not Reported | No | Yes | None |
| **17:** Ahmed *et al* | Not Reported | No | Yes | None |
| | Index Test | | | |
| | **Could the conduct or the interpretation of the test introduced bias?** | | | |
| Paper | SQ1 | SQ2 | Applicability | |
| **1:** Watany *et al* | Not Reported | No | No | |
| **2:** Agrawal *et al* | Not Reported | Not Applicable | No | |
| **3:** Bennet *et al* | Not Reported | No | No | |
| **4:** Bennet *et al* | Not Reported | No | No | |
| **5:** Bai *et al* | Not Reported | No | No | |
| **6:** Wen *et al* | Not Reported | No | No | |
| **7:** Choudhary *et al* | Not Reported | No | No | |
| **8:** Gopal *et al* | Not Reported | No | No | |
| **9:** Prasad *et al* | Not Reported | No | No | |
| **10:** Peng *et al* | Not Reported | No | No | |
| **11:** Ahmed *et al* | Not Reported | No | No | |
| **12:** Mousa *et al* | Not Reported | No | No | |
| **13:** Gopal *et al* | Not Reported | No | No | |
| **14:** Agrawal *et al* | Not Reported | No | No | |
| **15:** Nickavar *et al* | Not Reported | No | No | |
| **16:** Mishra *et al* | Not Reported | No | No | |
| **17:** Ahmed *et al* | Not Reported | No | No | |
| | Reference Standard | | | |
| | **Could the reference standard, its conduct or its interpretation have introduced bias? introduced bias?** | | | |
| Paper | SQ1 | SQ2 | Applicability | |
| **1:** Watany *et al* | Yes | Not Reported | No | |
| **2:** Agrawal *et al* | Yes | Not Reported | No | |
| **3:** Bennet *et al* | Yes | Not Reported | No | |

*(Continued)*

**Table 2.** (Continued)

| Paper | | | | |
|---|---|---|---|---|
| **4:** Bennet *et al* | Yes | Not Reported | No | |
| **5:** Bai *et al* | Yes | Not Reported | No | |
| **6:** Wen *et al* | Yes | Not Reported | No | |
| **7:** Choudhary *et al* | Yes | Not Reported | No | |
| **8:** Gopal *et al* | Yes | Not Reported | No | |
| **9:** Prasad *et al* | Yes | Not Reported | No | |
| **10:** Peng *et al* | Yes | Not Reported | No | |
| **11:** Ahmed *et al* | Yes | Not Reported | No | |
| **12:** Mousa *et al* | Yes | Not Reported | No | |
| **13:** Gopal *et al* | Yes | Not Reported | No | |
| **14:** Agrawal *et al* | Yes | Not Reported | No | |
| **15:** Nickavar *et al* | Yes | Not Reported | No | |
| **16:** Mishara *et al* | Yes | Not Reported | No | |
| **17:** Ahmed *et al* | Yes | Not Reported | No | |

| | **Flow and Timing** | | | **Overall risk of bias** |
|---|---|---|---|---|
| | **Could the patient flow have introduced bias?** | | | |
| Paper | SQ1 | SQ2 | SQ3 | |
| **1:** Watany *et al* | Not enough info | Yes | Yes | **Low Risk** |
| **2:** Agrawal *et al* | Yes | Yes | Yes | **Unknown Risk** |
| **3:** Bennet *et al* | Not enough info | Yes | Yes | **Low Risk** |
| **4:** Bennet *et al* | Not enough info | Yes | Yes | **Low Risk** |
| **5:** Bai *et al* | Not enough info | Yes | Yes | **Unknown Risk** |
| **6:** Wen *et al* | Yes | Yes | Yes | **Low Risk** |
| **7:** Choudhary *et al* | Not enough info | Yes | Yes | **Low Risk** |
| **8:** Gopal *et al* | Not enough info | Yes | Yes | **Low Risk** |
| **9:** Prasad *et al* | Yes | Yes | Yes | **Low Risk** |
| **10:** Peng *et al* | Not enough info | Yes | Yes | **Low Risk** |
| **11:** Ahmed *et al* | Yes | Yes | Yes | **Low Risk** |
| **12:** Mousa *et al* | Not enough info | Yes | Yes | **Low Risk** |
| **13:** Gopal *et al* | Yes | Yes | Yes | **Low Risk** |
| **14:** Agrawal *et al* | Yes | Yes | Yes | **Unknown Risk** |
| **15:** Nickavar *et al* | Yes | Yes | Yes | **Low Risk** |
| **16:** Mishara *et al* | Yes | Yes | Yes | **Low Risk** |
| **17:** Ahmed *et al* | Yes | Yes | Yes | **Low Risk** |

DOMAIN 1: PATIENT SELECTION Risk of bias: Could the selection of patients have introduced bias?

Signalling question 1: Was a consecutive or random sample of patients enrolled? Signalling question 2: Was a case-control design avoided? Signalling question 3: Did the study avoid inappropriate exclusions?

DOMAIN 2: INDEX TEST Risk of Bias: Could the conduct or interpretation of the index test have introduced bias?

Signalling question 1: Were the index test results interpreted without knowledge of the results of the reference standard? Signalling question 2: If a threshold was used, was it pre-specified?

DOMAIN 3: REFERENCE STANDARD Risk of Bias: Could the reference standard, its conduct, or its interpretation have introduced bias?

Signalling question 1: Is the reference standard likely to correctly classify the target condition? Signalling question 2: Were the reference standard results interpreted without knowledge of the results of the index test?

DOMAIN 4: FLOW AND TIMING Risk of Bias: Could the patient flow have introduced bias?

Signalling question 1: Was there an appropriate interval between index test and reference standard? Signalling question 2: Did all patients receive the same reference standard?

For each signalling question we have looked at the appropriate papers and assessed if they have answered the question. Responses that reduce the risk of bias have been marked in green while those that possibly increase bias have been marked in red. These have been taken together to assess the overall risk of bias.

kidney repair following kidney damage [42, 43]. One study, published in 2018, repurposed nephronectin as a marker of kidney repair during the early stages of corticosteroid treatment [23]. For these purposes nephronectin shows promise. However, patients will still be treated with steroids. They may well be removed if there is no evidence of repair (indicated by increased levels of nephronectin); however, ideally a biomarker for steroid resistance would be able to distinguish patients prior to treatment.

## Vitamin D Binding Protein (VDBP)

It has been found that vitamin D deficiency is associated to a greater degree with SRNS compared to SSNS [44]. It has been postulated that this marked vitamin D deficiency is due to the increased urinary loss of VDBP in SRNS versus SSNS [26]. VDBP is sufficiently small to pass through the glomerular filtration barrier. Proximal tubular cells reabsorb the lost VDBP via cubulin and megalin receptors. Hence chronic tubular injury could reduce this reabsorption leading to a greater loss of VDBP in the urine [26]. Hinting at the importance of VDBP as a biomarker of steroid resistance is the presence of VDBP, either on its own or as part of a panel, in four of the seventeen studies. Vitamin D Binding Protein (VDBP) is a circulating protein that binds to vitamin D to create a store of Vitamin D so that rapid vitamin D deficiency can be avoided [45]. Vitamin D deficiency is more pronounced in SRNS than in SSNS, and VDBP can be used to distinguish between these two conditions [29].

## Adiponectin (ADIPOQ)

ADIPOQ is a hormone, released by adipocytes, that helps to improve insulin sensitivity and is anti-inflammatory [46]. Low levels of adiponectin is correlated with albuminuria in mice and humans [47]. *ADIPOQ* knockout mice demonstrate significant podocyte injury and albuminuria. Adiponectin therapy in this model restores podocyte foot processes [48]. Elevated levels of serum adiponectin have been reported in patients with FSGS, chronic kidney disease, end-stage renal disease, those on dialysis and transplant recipients [49–51]. Total serum levels of adiponectin rise following the onset of nephrotic syndrome with notable changes in the ratios of the three adiponectin isoforms [52]. A recent study noticed that levels start lower and show a significant decrease following steroid treatment in children with SSNS whereas in children with SRNS levels start higher and increase following treatment [24]. It is under this context that Agrawal proposes using Adiponectin as an early indicator that steroids are working.

## Matrix Metalloproteinase-2 (MMP-2)

MMP-2 is a metalloproteinase that acts on collagen IV [53]. It is normally expressed by the mesangial cells in the glomerulus; however, during times of inflammation expression levels by the mesangial cells increase and podocytes begin to express MMP-2 [54]. Indeed, increased levels of MMP-2 in the sera have been seen in animal models of chronic kidney disease and in humans with chronic kidney disease [55–57]. MMP-2 can also activate MMP-1 and MMP-9 leading to further extracellular matrix remodelling (ECM) [58]. Increased MMP-2 in the serum and urine has been associated with progressive kidney fibrosis in chronic kidney disease [54, 59–62]. In children with SRNS there is a higher urinary MMP-2/creatinine ratio than in SDNS. This suggests that there may be ECM remodelling in both instances but that in SRNS there is a higher risk of renal fibrosis [63]. One study reported that MMP-2 was elevated in SSNS patients following treatment [24]. Again, this suggests that MMP-2 is useful as an early indicator that steroids may be working but does not help patients avoid steroid exposure altogether.

### Neutrophil Gelatinase-Associated Lipocalin (NGAL)

NGAL is a small 25kDa protein within the lipocalin family [64]. Though initially found in neutrophils, NGAL is expressed by many epithelial cells [65]. It has been widely shown that NGAL expression is upregulated following renal injury and as such is a powerful biomarker for AKI [66–69]. NGAL is a marker for chronic kidney disease progression and is significantly increased in patients with SRNS compared to those with SSNS (AUC0.91 p = <0.0001) [65]. However, it has also been found that calcineurin inhibitors, such as cyclosporine A, can increase NGAL levels [70].

### Fetuin-A

Fetuin-A is a carrier protein that has roles in insulin signalling and protease inhibition [71]. It is central to the pathogenesis of a myriad of conditions including insulin resistance, type 2 diabetes, metabolic disorders, cardiovascular disease and brain disorders [72–75]. In the kidney Fetuin-A protects the integrity of the tissues and levels drop dramatically as chronic kidney disease progresses [76]. Fetuin-A is significantly elevated in the urine during SRNS, suggesting a depletion in the serum leading to a lack of protease inhibition. This is an intriguing hypothesis since there is a body of work supporting the role of a circulating protease in idiopathic nephrotic syndrome [77, 78].

### Prealbumin

Prealbumin can be a sign of a hypercatabolic state often due to increased degradation of muscle mass [79].

### Acid Glycoprotein 1 (AGP-1)

AGP-1 is an acute phase protein released by hepatocytes in response to infection and inflammation [80]. It is generated from active vitamin D [81]. Urinary secretion of AGP-1 in healthy individuals is very low. However urinary secretion is detectable patients in a range of renal diseases including nephrotic syndrome [81].

### Alpha 1 Acid Glycoprotein 2 (AGP-2)

AGP-2 is very similar to AGP-1, with only 21 out of 181 amino acids being different [82]. AGP-1 isoforms outnumber AGP-2 by a ratio of 3:1 in the plasma under normal conditions, however this ratio is known to change in diseased states [83, 84].

### Alpha 2 Macroglobulin (A2MCG)

A2MCG accounts for 3–5% of plasma protein and is mainly synthesised in the liver [85]. It is a broad-spectrum protease which traps proteases in its "molecular cage" much like a Venus fly trap would trap its prey [86, 87]. A2MCG has a bait region which allows it to trap active proteases [88]. However, somewhat counterintuitively, A2MCG has been shown to enhance the activity of the protease thrombin by inhibiting the anticoagulant protein C/protein S system [89]. There is evidence that the circulating factor in INS is a protease, which makes A2MCGs inclusion here interesting [11, 77, 90].

### Thyroxine Binding Globulin (TBG)

TBG is a serine protease inhibitor, which again could have clear implications for the activity of the circulating factor if it is indeed a serine protease [91]. TBG is lost in the urine of nephrotic patients in sufficient quantities to cause subclinical or overt hypothyroidism [92].

### Alpha 1 B Glycoprotein (A1BG)

A1BG is a 54.3 kDa protein of unknown function [93, 94]. It has been found to be elevated in certain cancers [95].

### Haptoglobin

When erythrocytes are lysed the free heme group from haemoglobin can react with molecular oxygen to form superoxide. Haptoglobin binds haemoglobin in the circulation and protects tissues from oxidative damage [96]. Haptoglobin is mainly synthesised in the liver and the lungs, then secreted into the plasma [97]. In addition to its function as an antioxidant, haptoglobin also plays roles in angiogenesis, immunoregulation, the inhibition of nitric oxide and it stimulates tissue repair [98]. There is a significant increase in serum haptoglobin in SRNS patients versus SSNS patients [28].

### Urinary Protein Carbonyl Content (UPCC)

Chronic oxidative stress can result in systemic inflammation. In turn this can lead to secretion of pro-inflammatory cytokines and an exacerbation of proteinuria [99]. An imbalance between oxidants and anti-oxidants has long been known to exist in idiopathic nephrotic syndrome [100]. Since the oxidative stress is said to be higher in SRNS Vs SSNS then it stands to reason that there would be more UPCC [101], and this has been reported [35]. The pro-inflammatory cytokines induced in response to chronic oxidative stress can damage podocytes [102, 103].

### P-glycoprotein

P-glycoprotein is encoded by the *MDR-1* gene, it is a transporter that causes the efflux of toxins and drugs that are between 300 and 2000 kDa [104]. Cyclosporine is a P-glycoprotein antagonist that reduces the activity of the transporter allowing corticosteroids to accumulate within the cell to achieve a therapeutic effect [105]. The activity of P-glycoprotein was found to be significantly higher in the peripheral blood mononuclear cells (PBMCs) of steroid resistant patients compared to steroid sensitive patients [106]. P-glycoprotein expression is higher on the lymphocytes of steroid resistant patients versus steroid sensitive [107]. Additionally, glomerular expression of P-glycoprotein is significantly increased in those who frequently relapse and or steroid resistant or steroid dependent [108]. Genetic differences affecting the activity of P-glycoprotein can have an impact on the outcome of drug therapy [109] and a significant increase in the prevalence of a SNP in the *MDR1* gene (G2677T/A) amongst SRNS patients compared to SSNS has been reported [110].

### Multi Drug Resistance Protein 1 (MRP-1)

MRP-1 is pump responsible for the efflux of drugs acting on similar substrates to P-glycoprotein, but with a preference for heavy metal anions and toxins out of cells [111]. In contrast with most ABC transporters that are located on the apical membrane of cells and pump out into the urine or bile, MRP-1 is located on the basolateral membrane and pumps out into the interstitium [112, 113]. It is expressed throughout the body but particularly highly in PBMCs

and the kidney [114]. Increased expression of MRP-1 is known to be associated with SRNS and can be assayed for [31].

## Soluble Urokinase Plasminogen Activator Receptor (suPAR)

suPAR fulfils criteria of a circulating protein signalling to the kidney. It can readily be found in the circulation. It is generated by immature myeloid cells [14]. It can act directly on the podocyte via interaction with αvβIII integrin [115] or activates proximal tubular cell mitochondria [116]. The former has a deleterious effect on the podocytes, leading to foot process effacement and downregulation of podocin and nephrin [115]. This can affect the structure and function of the glomerulus [117–120]. Urinary suPAR is known to increase in FSGS and positively correlates with disease severity [121, 122]. Additionally it has been shown to be able to predict recurrence of disease in transplanted patients [123]. Urinary and serum levels of suPAR can also stratify cases of minimal change disease and FSGS [124]. suPAR has been hypothesised to be a circulating factor driving the pathogenesis of nephrotic syndrome. However, there is controversy over the role of suPAR as a circulating factor, with many studies not corroborating the original reports, and showing highly variable levels in cohorts of NS patients [125–128]. Moreover, suPAR is a known inflammatory mediator and it has been associated with other conditions [129]. Notably suPAR levels can be influenced by diabetes and obesity [130, 131].

## Interleukin-7 (IL-7)

IL-7 is secreted by T cells. T cells were postulated to be the source of the illusive circulating factor driving INS almost fifty years ago [132]. IL-7 is a cytokine that supports host defence by regulating the homeostasis of the cells of the immune system such that congenital deficiency of IL-7 leads to severe immunodeficiency [133]. In the mouse model of Adriamycin nephropathy, IL-7 has been shown to lead to impaired barrier function, podocyte apoptosis, impaired activation of nephrin and actin cytoskeleton dysregulation [134].

## Interleukin-9 (IL-9)

IL-9, also released by T cells, seems to act antagonistically to IL-7 on the podocyte in nephrotic syndrome pathogenesis. IL-9 can dramatically improve glomerular function in the Adriamycin nephropathy model. However, it is known to increase in the serum of patients with primary FSGS hence it's utility here as a biomarker [135].

## Interleukin-8 (IL-8)

IL-8 is released by T-cells or resident kidney cells in response to pro-inflammatory stimuli [136]. Within the kidney IL-8 is produced by mesangial cells [137], podocytes [138] and tubular epithelial cells [139]. IL-8 is known to affect the functioning of the glomerular basement membrane [140]. It has been shown in rats that IL-8 treatment decreases the synthesis of heparin sulfate proteoglycans leading to proteinuria [141].

## Monocyte Chemoattractant Protein-1 (MCP-1)

MCP-1 is a chemokine that recruits monocytes from the bone marrow to sites of inflammation [142]. There is a significant increase in the infiltration and accumulation of macrophages in the glomeruli of children with SRNS versus SSNS. Interestingly, there is an increase of MCP-1 in the urine but not in the serum of FSGS patients which suggests that the MCP-1 is being produced by the kidney [143]. It is known that the mesangial [144] cells of the glomerulus can

produce MCP-1 [145–147]. Persistently elevated levels of MCP-1 in the urine could demonstrate continuing inflammation within the kidney and a resistance to steroid treatment [148].

### 50S ribosomal protein L32 (50SL32) and 30S ribosomal protein S11 (30SS11)

50SL32 and 30SS11 are both subunit proteins that make up ribosomes.

### S-adenosylmethionine decarboxylase α chain (SAMDC)

SAMDC is a critical enzyme involved in the synthesis of polyamines. It is an amino acid decarboxylase that is essential to life [149].

### FK506-binding protein 1A (FKBP12)

FKBP12 is the target for the immunosuppressant drug tacrolimus, also known as FK506. FK506 binds to FKBP12 leading to the inhibition of calcineurin which has a frontline treatment for nephrotic syndrome [150, 151]. Within the glomeruli of the kidney FKBP12 is expressed exclusively by the podocytes [152].

Table 3 shows the identified candidate biomarkers separated by source and whether they are predictive or evaluative.

Created by author. The biomarkers were ranked according to their sample size which ranged from 18 participants for a study of the N-Acetyl-beta-D Glucosaminodase (NAG)/creatinine ratio to 254 participants for a study on P-glycoprotein and MRP-1. The biomarkers were also ranked by their BIOCROSS score giving the lowest scoring paper (candidate 5 [27]) point and the highest scoring paper (candidate 1 [23]) 17 points. The full data is shown in Table 3. Additionally, biomarkers were scored by their sensitivity/specificity according to the ranges they fell within (Fig 2). Papers that did not disclose sensitivity/specificity values received 0 points. The candidate biomarkers have been separated according to whether they are predictive or evaluative and which biological specimen they can be found in.

## Discussion

The field of clinical nephrology is working toward finding predictive biomarkers for SRNS to save patients being exposed to futile steroid treatments. Unfortunately for these kinds of studies INS as a whole is rare and SRNS even more so. The importance of undertaking studies with an adequate sample size is demonstrated by the two studies for candidate biomarkers 12 and 15 [34, 37]. Both reported 100% diagnostic accuracy for the candidate markers under investigation (serum suPAR as an evaluative marker and NGAL/creatinine ratio as a predictive urinary marker). These molecules need now to be evaluated prospectively. It was recently done for suPAR in the prediction of outcomes in septic acute kidney injury [153].

The search terms were set up to very specifically identify studies looking at SRNS. The terminology used by nephrologists and renal scientists is a challenge. Focal segmental glomerulosclerosis (FSGS) is often but not always steroid resistant. Equally SRNS usually presents histopathologically as FSGS, but again, not always. This has led to some using the terms FSGS and SRNS interchangeably. SRNS is a clinical phenotype, indeed it is the key characteristic of interest in this systematic review. Hence SRNS was prioritised in the search strategy. This will have biased specificity at the expense of selectivity in the identified studies and biomarkers. We are satisfied that this sacrifice was necessary and will have led to the identification of credible candidate biomarkers for future analysis.

**Table 3. Candidate biomarker ranking.**

| Rank | Candidate | Biomarker | Paper | Sample Size Score | BIOCROSS Rank Score | Sensitivity/ Specificity Rank | Sum |
|---|---|---|---|---|---|---|---|
| | | **Predictive Urinary Biomarkers** | | | | | |
| 1 | 7 | VDBP and NGAL | Choudhary *et al* 2020 | 10 | 13 | 4 | 27 |
| 2 | 13 | Urinary Protein Bound Sialic Acid | Gopal *et al* 2016 | 13 | 10 | 2 | 25 |
| 3 | 8 | UPCC | Gopal *et al* 2017 | 12 | 4 | 6 | 22 |
| 4 | 16 | N-Acetyl-beta-D Glucosaminodase /creatinine ratio | Mishra *et al* 2012 | 1 | 10 | 10 | 21 |
| 5 | 17 | IL-8 | Ahmed *et al* 2019 | 11 | 2 | 6 | 19 |
| 6 | 15 | NGAL/Creatinine ratio | Nickavar *et al* 2016 | 5 | 4 | 10 | 19 |
| Rank | Candidate | Biomarker | Paper | Sample Size Score | BIOCROSS Rank Score | Sensitivity/ Specificity Rank | Sum |
| | | **Predictive Serum/Plasma Biomarkers** | | | | | |
| 1 | 6 | Haptoglobin | Wen *et al* 2012 | 15 | 13 | 10 | 38 |
| 2 | 10 | suPAR | Peng *et al* 2015 | 16 | 15 | 2 | 33 |
| 3 | 2 | A panel of VDBP, ADIPOQ and MMP-2 | Agrawal *et al* 2020 | 2 | 10 | 0 | 12 |
| 4 | 14 | IL-7, IL-9 and MCP-1 | Agrawal *et al* 2021 | 3 | 2 | 2 | 7 |
| Rank | Candidate | Biomarker | Paper | Sample Size Score | BIOCROSS Rank Score | Sensitivity/ Specificity Rank | Sum |
| | | **Evaluative Urinary Biomarkers** | | | | | |
| 1 | 4 | VDBP | Bennet *et al* 2016 | 8 | 4 | 4 | 16 |
| 2 | 3 | A panel of VDBP, NGAL, Fetuin-1, Prealbumin, Alpha-1-Acid Glycoprotein 2, AGP1, A2MCG, A1BG, TBG and Hemopexin | Bennet *et al* 2017 | 6 | 4 | 4 | 14 |
| 3 | 3 | A panel of VDBP, NGAL, Fetuin-1, Prealbumin, Alpha-1-Acid Glycoprotein 2, AGP1, A2MCG, A1BG, TBG and Hemopexin | Bennet *et al* 2017 | 6 | 4 | 4 | 14 |
| 4 | 5 | A panel of 50SL32, SAMDC, FKBP12 and 30SS11 | Bai *et al* 2013 | 4 | 1 | 8 | 13 |
| Rank | Candidate | Biomarker | Paper | Sample Size Score | BIOCROSS Rank Score | Sensitivity/ Specificity Rank | Sum |
| | | **Evaluative Serum/Plasma Biomarkers** | | | | | |
| 1 | 1 | Nephronectin | Watany *et al*,2018 | 14 | 17 | 6 | 37 |
| 2 | 12 | suPAR | Mousa *et al* 2018 | 7 | 4 | 10 | 21 |
| 3 | 11 | Endothelin-1 | Ahmed *et al* 2019 | 9 | 4 | 8 | 21 |
| Rank | Candidate | Biomarker | Paper | Sample Size Score | BIOCROSS Rank Score | Sensitivity/ Specificity Rank | Sum |
| | | **Evaluative PBMC Biomarkers** | | | | | |
| 1 | 9 | P-Glycoprotein and MRP-1 | Prasad *et al* 2021 | 17 | 15 | 6 | 38 |

We did not set a lower limit for sample size. Nephrotic syndrome and its subdivisions are rare diseases hence recruitment can be difficult. We accept that smaller sample sizes of patient groups are likely to be less representative and more prone to error however, we wanted to generate a list of candidate biomarkers in the field for further validation.

| 1 | Nephronectin | 7 | Vitamin D Binding Protein and Neutrophil Gelatinase-Associated Lipocalin | 13 | Urinary Protein Bound Sialic Acid |
|---|---|---|---|---|---|
| 2 | Vitamin D Binding Protein, Adiponectin and Metalloproteinase-2 | 8 | Urinary Protein Carbonyl Content | 14 | Interleukin-7, Interleukin-9 and Monocyte Chemoattractant Protein-1 |
| 3 | Vitamin D Binding Protein, Neutrophil Gelatinase-Associated Lipocalin, Fetuin-1, Prealbumin, Alpha-1-Acid Glycoprotein 2, AGP1, Alpha-2-Macroglobulin, Alpha-1-B Glycoprotein, Thyroxine-Binding Globulin and Hemopexin | 9 | P-Glycoprotein | 15 | Neutrophil Gelatinase-Associated Lipocalin /Creatinine |
| 4 | Vitamin D Binding Protein | 10 | Soluble Urokinase Plasminogen Activator Receptor | 16 | N-Acetyl-beta-D Glucosaminodase /Creatinine |
| 5 | 50S Ribosomal Protein L32, S-Adenosylmethionine Decarboxylase α Chain, FK506-Binding Protein 1A and 30S ribosomal protein S11 | 11 | Endothelin-1 | 17 | Interleukin-8 |
| 6 | Haptoglobin | 12 | Soluble Urokinase Plasminogen Activator Receptor | | |

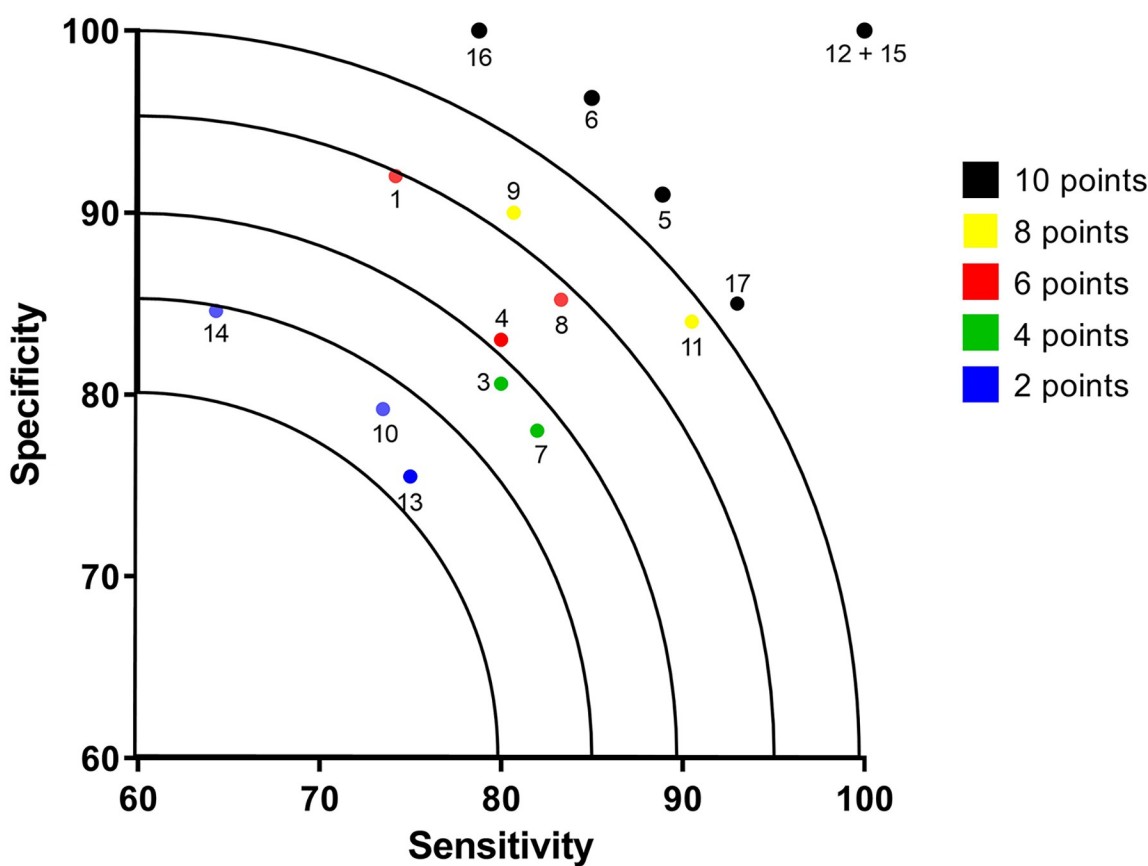

**Fig 2. Specificity and sensitivity of candidate biomarkers and panels.**

Owing to the scarcity of NS patients some of the studies incorporated here were cross-sectional. This has meant that the cohorts in these studies incorporate a mixture of steroid naïve, those who are currently on steroids and those who have met diagnostic criteria for being either steroid sensitive or steroid resistant. Comparisons between these groups must be done with great caution. Many of the studies in this review would have been stronger if they had more clearly explained why they chose a cross-sectional design, and if they had taken steps to reduce the influence of confounding variables on their results.

While some of the presented studies controlled for the exposure of subjects to immunosuppressive treatment, many did not. It is vital when looking for biomarkers of steroid resistance to receive sufficient information concerning treatment to understand the context of the biomarkers in those subjects. Given the paucity of information in many of the identified studies concerning the treatment regimen, the authors recommend that future studies pay particular attention to this aspect of their biomarker studies.

Overall, studies were considered to have low risk of bias. Three studies were rated as having an unknown risk of bias [24, 27, 36], which was because of an inadequate description of the exclusion criteria, and as such it is impossible to determine how generalisable their study group is and how well the studies addressed potential confounding factors.

The main limitations of studies reported to date include sampling method, study design, application of cut-off threshold, investigator blinding, and timing of testing.

None of the included studies reported a sampling method for patient recruitment. Given the rarity of INS it is likely that studies used consecutive rather than random sampling to recruit patients, but this is not evaluable. Choice of study design is also a concern. Most identified in this review used a case-control design. It is preferable to avoid case-controlled study design, because it can be difficult to identify an appropriate control group to reduce risk of bias [154]. However, such designs have practical advantage for recruiting cases when the disease is rare. The reporting of control selection is therefore critical to assess risk of bias.

None of the selected papers pre-specified the threshold cut-off. By selecting the cut-off after the analysis, the data is shown in its best light and is therefore likely to lead to an overestimation of the abilities of the index test. Since all the papers that used a cut-off failed to pre-specify, they remain comparable. However, it is worth pointing out that the same cut-offs would be unlikely to yield the same sensitivity/specificity values in a new cohort. Where possible, authors should pre-specify the cut-offs to increase the accuracy of their sensitivities and specificities.

Future studies are also encouraged to undertake assessor blinding, which enables the impartial analysis of the index test. None of the studies included in this review provided any details about blinding of the results of the index test of reference standard.

It is also important to report when the index test was performed relative to when the samples were taken. Progression of the pathophysiology will have an impact on the abundance of the biomarkers being tested. Again, to make an accurate assessment of risk of bias studies need to be given all the information. Almost half of studies included in this review (8/17) did not report any information regarding when the index test took place relative to the reference standard.

To improve the clinical management of patients with steroid resistant nephrotic syndrome, the goal is to identify predictive markers that will obviate the need to expose these patients to toxic, ineffective treatment.

Candidate biomarkers identified in this review were ranked according to a combination of their sample size, the BIOCROSS score and their sensitivity/specificity. Though P-glycoprotein and MRP-1 scored the most according to these criteria, it is less clinically useful since these are evaluative rather than predictive biomarkers. Haptoglobin was the most rigorously tested

predictive marker with the most promising sensitivity/specificity values, closely followed by suPAR.

The biggest flaw with the current evidence-base is how reliant these biomarker studies are on case-controlled studies. Many of the limitations of the studies included here could be overcome by adopting the PRoBE (Prospective-Specimen Collection Retrospective Blinded Evaluation) approach using a biobank such as NURTuRE-INS, which currently contains samples from 742 INS patients [155, 156]. NURTuRE-INS is a well-defined prospective cohort that collects blood and urine from both steroid resistant and steroid sensitive idiopathic nephrotic syndrome patients. The bio samples are collected during periods of active disease relapse) and remission providing vital internal controls for each patient. Despite the multi-centre approach, samples are all handled to the same exacting protocol. The 23 renal centres across the UK collect, process, and freeze the samples at -80°C within 2 hours. There is a chronic kidney disease arm of NURTuRE that could provide an additional source of control samples. These could control for markers of inflammatory processes common to multiple kidney diseases.

The PRoBE approach deals with several biases that can be inherent with retrospective case-control study designs. Spectrum bias occurs when case-patients with clear cut examples of the disease (usually severe and/or well-documented) are compared with carefully selected, particularly healthy controls [157]. The manuscripts selected for review here did not describe how patients were sampled (e.g., consecutive, or random), without such information it is difficult to judge the risk of spectrum bias. However, this bias can be avoided when using the PRoBE approach. Subjects in the cohort are identified as patients or controls, then the study group is randomly selected from these sub-groups.

Additionally, by using nested subgroups the discovery and evaluation phases of biomarker identification can be carried out in the same population [156].

suPAR and haptoglobin have emerged from this systematic review as the most promising biomarkers for the prospective distinction between steroid resistant and steroid sensitive variants of idiopathic nephrotic syndrome.

Haptoglobin is known to be an acute phase protein which is elevated in many inflammatory diseases and helps to coordinate the immune response [158]. Whilst the role, if indeed it has one, in NS pathogenesis is yet to be elucidated [159], it is known to regulate the function of lymphocytes and macrophages and control tissue damage in the context of inflammation [160]. Therefore, it is logical that haptoglobin could indicate steroid responsiveness.

suPAR is known to exert direct effects on the podocyte and has been shown to downregulate nephrin and podocin via activation of the αβV III integrin [161, 162]. When cultured podocytes are treated with suPAR they respond by upregulating expression of TRPC6, which can also be achieved by treating blood samples from FSGS patients [163]. This further underpins a role for TRPC6 in the pathogenesis of nephrotic syndrome [90]. It has been suggested that suPAR could be the putative circulating factor in INS [164, 165]. Though this has been disputed [166, 167]. However, in addition to its possible role as a biomarker for steroid responsiveness, urinary suPAR has shown utility in predicting recurrence of FSGS following a kidney transplant [122].

It is our strong recommendation that work continues to investigate the utility of these markers using the PRoBE approach on a cohort such as NURTuRE-INS.

## Supporting information

**S1 Checklist. PRISMA 2020 checklist.**
(DOCX)

**S1 Table. Literature search results.** Supplementary table one shows all of the papers identified by our search strategy and the reason for their exclusion (if applicable).
(XLSX)

## Author Contributions

**Conceptualization:** Carl J. May.

**Data curation:** Carl J. May.

**Formal analysis:** Carl J. May.

**Investigation:** Carl J. May.

**Supervision:** Nathan P. Ford.

**Writing – original draft:** Carl J. May.

**Writing – review & editing:** Nathan P. Ford, Gavin I. Welsh, Moin A. Saleem.

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
