## [Decision Letter · Decision Letter 0]

10 Jul 2023

PONE-D-23-12560Biomarkers to predict steroid resistance in idiopathic nephrotic syndrome: a systematic review.PLOS ONE

Dear Dr. May,

Thank you for submitting your manuscript to PLOS ONE. After careful consideration, we feel that it has merit but does not fully meet PLOS ONE’s publication criteria as it currently stands. Therefore, we invite you to submit a revised version of the manuscript that addresses the points raised during the review process.

We look forward to receiving your revised manuscript.

Kind regards,

Keiko Hosohata, Ph.D.

Academic Editor

PLOS ONE

Journal Requirements:

2. Please ensure that you refer to Figure 3 and 5 in your text as, if accepted, production will need this reference to link the reader to the figure.

Reviewers' comments:

Reviewer's Responses to Questions

**Comments to the Author**

1. Is the manuscript technically sound, and do the data support the conclusions?

Reviewer #1: No

Reviewer #2: No

Reviewer #3: Partly

2. Has the statistical analysis been performed appropriately and rigorously? 

Reviewer #1: Yes

Reviewer #2: N/A

Reviewer #3: Yes

3. Have the authors made all data underlying the findings in their manuscript fully available?

Reviewer #1: Yes

Reviewer #2: Yes

Reviewer #3: Yes

4. Is the manuscript presented in an intelligible fashion and written in standard English?

Reviewer #1: Yes

Reviewer #2: Yes

Reviewer #3: Yes

5. Review Comments to the Author

Reviewer #1: This is a timely review of potential biomarkers to predict or evaluate steroid resistance in NS. Its main strength is the rigor applied to analyzing the available publications on this topic. However, numerous details presented would need to be clarified and reorganized in a more readily interpretable format to enhance the readability and usefulness of the manuscript. Suggested edits are detailed below, but are listed by section, since page and line numbers were not included.

TITLE – The title denotes predictive markers but the data presented shows both predictive and evaluative markers… please edit title to more accurately reflect the data you are presenting.

METHODS

Figure 2 - Reference to Figure 1 here seems incorrect. Please add details about the BIOCROSS methodology and scoring metrics to improve readability.

Figure 3 – Please add details about this assessment here as well.

Figure 4 – Please add details here as well… The readers cannot be expected to be familiar with your methods…

Listing of individual candidate biomarkers –

Please either list each marker in rank order in the text based on your criteria for usefulness, or at least group them into 2-4 tiers in order of strength of evidence to help the readers understand the final results of your detailed analyses.

Please clearly classify all markers throughout the text as either serum, plasma, or urine markers. Indeed, the manuscript could arguably be more useful to readers if it were organized by serum vs. plasma vs. urine biomarkers.

VDBP paragraphs appear twice and should be merged…

Some markers discussed (Fetuin-A, AGP, etc…) do not seem to even be on your list of 17 in Figure 2. Please correct and/or reconcile this.

AGP-1 needs to be defined.

MRP-1 needs to be defined.

Some references appear to be incorrect. Please verify the accuracy of all references throughout the manuscript.

Figure 3 – Please add a column at left side to name each biomarker or panel to enhance readability of the table. Also, the text below “Reference Standard” seems incorrect and needs editing.

Figure 4 – This figure nicely separates the markers by quality of evidence but is uninterpretable without the addition of the actual names of the markers on the figure… Please add individual names instead of numbers that require the reader to find another table where they were numbered. Also, please change both axes to stop at 100%. A potential alternative would be to create a table of the biomarkers that is organized by quartiles based on the strength of evidence.

Figure 5 – Please consider separating the markers in this table based on predictive vs. evaluative, and list them in order of strength of evidence to enhance the interpretability for the readers.

Reviewer #2: In this manuscript, authors summarized 17 papers published between 01/01/2012 and 10/05/2022 concerning biomarkers that can distinguish between SSNS and SRNS. Haptoglobin and suPAR were identified as the most promising biomarkers in this study.

There are also some problems existed in this manuscript as follows:

1. The function of each biomarker on the kidney is described at length in the results section. Finally, suPAR and haptoglobin were selected as the most promising biomarkers for the distinction between SSNS and SRNS. However, in the discussion section, their importance is not highlighted. It is inappropriate.

2. There are some linguistic gleanings that require a careful revision. The paper needs to be edit with the English editing companies.

3. the paper is not well-written, figures are vague and hardly to read.

Reviewer #3: Nephrotic syndrome is a rare disease with an important impact, especially on children's lives. Actually, it is still unknown what the cause of INS is, and it is not possible to know if a patient will respond or Nenot  to steroid treatment. This is important to avoid side effects and useless therapies.

The authors made a review in this way: which biomarker (if any) is able to distinguish steroid responders from no responders?

Unfortunately, this review has several issues.

This review is mainly a list of biomolecules; it is not a negative point, but often it lacks information and it is not clear how the molecules are listed. I suggest to the authors to separate the biomarkers into groups like useful as early treatment, useful prior to treatment, and not useful (like AGP1, which seems able only to identify INS).

The title of each molecule could be uniform. Actually, some of the themes are presented as acronyms and others as full names. I suggest writing with your full name and acronym.

Moreover, VDBP is presented twice; please fuse them into one paragraph.

It is important for each molecule to report the obtained results from considered studies and the obtained score at your analysis.

As the authors wrote regarding MDR1’s SNP, it could also be useful to write about other possible SNPs able to distinguish SR from no responder.

6. PLOS authors have the option to publish the peer review history of their article (what does this mean?). If published, this will include your full peer review and any attached files.

Reviewer #1: No

Reviewer #2: No

Reviewer #3: No

---

## [Author Response · Author response to Decision Letter 0]

5 Sep 2023

Reviewer #1 Comments

TITLE – The title denotes predictive markers but the data presented shows both predictive and evaluative markers… please edit title to more accurately reflect the data you are presenting.

• I have amended the title from “Biomarkers to predict steroid resistance in idiopathic nephrotic syndrome: a systematic review.” to “Biomarkers to predict or measure steroid resistance in idiopathic nephrotic syndrome: a systematic review.”. 

Figure 2 - Reference to Figure 1 here seems incorrect. 

• I erroneously referred to figure 2 instead of figure 1. This has now been amended

Please add details about the BIOCROSS methodology and scoring metrics to improve readability.

• Thank you for pointing this out, we have added the following paragraph to the methods section: “BIOCROSS is a quality assessment tool that is used to quantify the quality of the data that supports candidate biomarkers. The methodology of this tool is covered including details of the scoring is covered here . Briefly, the tool includes 10-items covering 5 domains: ‘Study rational’, ‘Design/Methods’, ‘Data analysis’, ‘Data interpretation’ and ‘Biomarker measurement’, aiming to assess different quality features of biomarker cross-sectional studies.”

Figure 3 – Please add details about this assessment here as well.

• Again, thanks for pointing this out, we have added the following: “The QUADAS-2 tool assesses the design and publication of biomarker data for applicability and risk of bias across four domains: patient selection, index test, reference standard, and flow and timing. It allows researchers to compare the risk of bias across different studies. The specific methodology is covered in detail here.”

Details of the domains and their respective signalling questions have been added to the manuscript as follows: 

“DOMAIN 1: PATIENT SELECTION Risk of bias: Could the selection of patients have introduced bias? 

Signalling question 1: Was a consecutive or random sample of patients enrolled? Signalling question 2: Was a case-control design avoided? Signalling question 3: Did the study avoid inappropriate exclusions?

DOMAIN 2: INDEX TEST Risk of Bias: Could the conduct or interpretation of the index test have introduced bias? 

Signalling question 1: Were the index test results interpreted without knowledge of the results of the reference standard? Signalling question 2: If a threshold was used, was it pre-specified?

DOMAIN 3: REFERENCE STANDARD Risk of Bias: Could the reference standard, its conduct, or its interpretation have introduced bias? 

Signalling question 1: Is the reference standard likely to correctly classify the target condition? Signalling question 2: Were the reference standard results interpreted without knowledge of the results of the index test?

DOMAIN 4: FLOW AND TIMING Risk of Bias: Could the patient flow have introduced bias? 

Signalling question 1: Was there an appropriate interval between index test and reference standard? Signalling question 2: Did all patients receive the same reference standard?”

Figure 4 – Please add details here as well… The readers cannot be expected to be familiar with your methods…

• Figure 4 shows a plot of sensitivity vs specificity, We feel the figure legend is sufficient as this is a simple synthesis. “Each biomarker or panel had its specificity plotted against its sensitivity. Points were awarded according to the increasing level of specificity and sensitivity.”

Please either list each marker in rank order in the text based on your criteria for usefulness, or at least group them into 2-4 tiers in order of strength of evidence to help the readers understand the final results of your detailed analyses.

• We appreciate your point of view. The order of the biomarkers presented in figures 2 and 5 is the same as the order that they were presented following the original literature search. We have added a sentence to the figure legends of figure 2 and figure 5 to make this clear, “candidate biomarkers are presented in the same order as revealed by the literature search”. We agree that 2-4 tiers in order of strength of evidence may well help the reader. However, the boundaries of these tiers would be somewhat arbitrary. As such we may unduly influence the reader as to the validity of each biomarker. As the review stands we are laying the evidence bare and are objectively scoring the biomarkers and their respective source manuscripts based on published quality assessment tools. Respectfully, we have left the tables presented in figures 2 and 5 as they are in terms of the order of the presented biomarkers. 

Please clearly classify all markers throughout the text as either serum, plasma, or urine markers. Indeed, the manuscript could arguably be more useful to readers if it were organized by serum vs. plasma vs. urine biomarkers.

• There is a column in Figure 2 that indicates the source of the biomarker, for increased clarity this column has been added to the table shown in Figure 5. We feel that subdividing the biomarkers based on their source would make comparing the relative performance of each candidate more difficult

VDBP paragraphs appear twice and should be merged…

• Thank you, these have been merged. 

Some markers discussed (Fetuin-A, AGP, etc…) do not seem to even be on your list of 17 in Figure 2. Please correct and/or reconcile this.

• Fetuin-A and AGP are present in the panels of biomarkers. We have now made sure that all biomarkers from both panels are covered. Additionally we have made it clear in the prose that the some of the biomarkers are effective as part of a panel rather than in their own right. 

AGP-1 needs to be defined.

MRP-1 needs to be defined.

• These definitions have now been added.

Some references appear to be incorrect. Please verify the accuracy of all references throughout the manuscript.

• Thanks for pointing this out. We have now manually verified ech reference individually and are satisfied that they are accurate.

Figure 3 – Please add a column at left side to name each biomarker or panel to enhance readability of the table. Also, the text below “Reference Standard” seems incorrect and needs editing.

• Adding the full name of each biomarker would make the tables cumbersome and could well reduce the readability of the tables. This information has been included elsewhere and is possibly tautologous to include here too. However, we totally agree that naming the biomarkers that comprise the panel’s would be useful here and so we have added this to Figure 2 and Figure 5

Figure 4 – This figure nicely separates the markers by quality of evidence but is uninterpretable without the addition of the actual names of the markers on the figure… Please add individual names instead of numbers that require the reader to find another table where they were numbered. Also, please change both axes to stop at 100%. A potential alternative would be to create a table of the biomarkers that is organized by quartiles based on the strength of evidence.

• We appreciate and understand your input. We have kept the numbers in the graph to stop it becoming too confusing, but have added a key so that the numbers can be more easily interpreted without searching through the manuscript. We have limited each axis to 100%

Figure 5 – Please consider separating the markers in this table based on predictive vs. evaluative, and list them in order of strength of evidence to enhance the interpretability for the readers.

• Thank you for pointing this out. We have rearranged the layout of the table shown in Figure 5 to show predictive and evaluative markers listed in order of strength of evidence.

Reviewer #2 Comments

The function of each biomarker on the kidney is described at length in the results section. Finally, suPAR and haptoglobin were selected as the most promising biomarkers for the distinction between SSNS and SRNS. However, in the discussion section, their importance is not highlighted. It is inappropriate.

• We apologise for this and have now added some information towards the end of the discussion section.

There are some linguistic gleanings that require a careful revision. The paper needs to be edit with the English editing companies.

• This was a criticism not identified by the other reviewers, nevertheless I have enlisted the help of a couple of native English-speakers (like me) to proof-read the manuscript.

the paper is not well-written, figures are vague and hardly to read.

• I am at a bit of a loss as to how to address this criticism save for re-reading the manuscript with this viewpoint in mind. Hopefully by addressing all the comments described by yourself and the other reviewers you will find the revised manuscript much improved. 

Reviewer #3 Comments

This review is mainly a list of biomolecules; it is not a negative point, but often it lacks information, and it is not clear how the molecules are listed. I suggest to the authors to separate the biomarkers into groups like useful as early treatment, useful prior to treatment, and not useful (like AGP1, which seems able only to identify INS).

• Thank you for your comment. Indeed, reviewer 1 identified a similar issue. We have now made clear that in Figure 2 the biomarkers are listed in the order they appeared when we performed the literature search, and in Figure 5 the biomarkers are now listed by category in terms of whether they are predictive or evaluative. I am not sure I understand your point regarding AGP-1. AGP-1 seems to be a marker of kidney lesions and has been found to be elevated in a range of kidney diseases. It is included in this systematic review as part of the panel of 10 biomarkers published by Bennet et al in 2017

The title of each molecule could be uniform. Actually, some of the themes are presented as acronyms and others as full names. I suggest writing with your full name and acronym.

• You are correct. We have now amended the manuscript in response to this recommendation. The tables in Figures 2 and 5 now have the name of the biomarker and, where appropriate, the acronym. In the prose, each biomarker is introduced by its full name and then the acronym underneath. We hope this increases clarity for the reader.

VDBP is presented twice; please fuse them into one paragraph.

• This has now been rectified.

It is important for each molecule to report the obtained results from considered studies and the obtained score at your analysis.

• The scores calculated and shown in Figure 5 are based on each candidate biomarker or panel of biomarkers. The scores are laid out in separate columns along with a sum of the scores.

As the authors wrote regarding MDR1’s SNP, it could also be useful to write about other possible SNPs able to distinguish SR from no responder.

• We totally agree. However, due to the way systematic reviews are performed we are only able to include manuscripts that were identified following the literature search and according to the inclusion and exclusion criteria. We will keep this in mind for future work as genotyping tools would be very valuable.

---

## [Decision Letter · Decision Letter 1]

26 Feb 2024

PONE-D-23-12560R1Biomarkers to predict or measure steroid resistance in idiopathic nephrotic syndrome: a systematic review.PLOS ONE

Dear Dr. May,

Thank you for submitting your manuscript to PLOS ONE. After careful consideration, we feel that it has merit but does not fully meet PLOS ONE’s publication criteria as it currently stands. Therefore, we invite you to submit a revised version of the manuscript that addresses the points raised during the review process.

A rebuttal letter that responds to each point raised by the academic editor and reviewer(s). You should upload this letter as a separate file labeled 'Response to Reviewers'.A marked-up copy of your manuscript that highlights changes made to the original version. You should upload this as a separate file labeled 'Revised Manuscript with Track Changes'.An unmarked version of your revised paper without tracked changes. You should upload this as a separate file labeled 'Manuscript'

We look forward to receiving your revised manuscript.

Kind regards,

Rajendra Bhimma, PhD

Academic Editor

PLOS ONE

Journal Requirements:

Additional Editor Comments:

See comments by reviewers.

Reviewers' comments:

Reviewer's Responses to Questions

**Comments to the Author**

1. If the authors have adequately addressed your comments raised in a previous round of review and you feel that this manuscript is now acceptable for publication, you may indicate that here to bypass the “Comments to the Author” section, enter your conflict of interest statement in the “Confidential to Editor” section, and submit your "Accept" recommendation.

Reviewer #1: (No Response)

2. Is the manuscript technically sound, and do the data support the conclusions?

Reviewer #1: Partly

3. Has the statistical analysis been performed appropriately and rigorously? 

Reviewer #1: Yes

4. Have the authors made all data underlying the findings in their manuscript fully available?

Reviewer #1: Yes

5. Is the manuscript presented in an intelligible fashion and written in standard English?

Reviewer #1: Yes

6. Review Comments to the Author

Reviewer #1: This revised manuscript seems improved, but the authors appear to have only been moderately responsive to the Reviewers' prior concerns and requests.

I agree with Reviewer 1's concerns about creating a ranking or at least a 2-4 tier ranking of the identified biomarkers to better enable readers to benefit from and potentially make use of the detailed analyses done by the authors.

I also agree with Reviewer 1's request to present the biomarkers by BOTH this ranking AND separated by their source material, as the potential future utility (and commercial viability) of the identified biomarkers may be different depending on their source.

This manuscript describes a nice approach to evaluate biomarkers to distinguish SRNS and SSNS. It could have benefitted however from the inclusion of a clinical nephrologist as a co-author both to improve the readability and clinical relevance of the identified findings. It also should be submitted double-spaced and include page numbers and line numbers to facilitate review of the manuscript. Several other specific opportunities to improve the manuscript are noted below.

1 - The authors need to add a paragraph to the discussion addressing the potential impact of confounding of the results presented in their study due to the presence vs. absence of ongoing immunosuppressive treatment of NS on the reported values for biomarkers of NS, as few of the reported studies controlled for the presence vs. absence of immunosuppressive treatment, or even addressed this concern as a major potential source of confounding of their data.

2 - Please add more details about the NS subgroups described midway through the Introduction, and their relative risks for progression to kidney failure.

3 - Please add the details promised to Reviewer 1 in your rebuttal letter to the Methods section.

4 - In Figure 1, please add details about the ranges and interpretation of the BIOCROSS scores in the table to improve reader interpretability.

5 - Figure 2 is too small for the readers to see. Please either enlarge it to make it readable or consider omitting.

6 - Figure 3 should include the names of each dot directly in the figure, and the legend should precisely clarify the meaning of the QUADRAS scores shown in the figure to improve readability.

7 - As also promised in the rebuttal letter, please spell out the full names of all biomarkers followed by the acronyms after their first usage consistently throughout the manuscript.

8 - For Figure 4, I suggest the same improvements as for Figure 3.

7. PLOS authors have the option to publish the peer review history of their article (what does this mean?). If published, this will include your full peer review and any attached files.

Reviewer #1: No

---

## [Author Response · Author response to Decision Letter 1]

18 Sep 2024

Reviewer #1: This revised manuscript seems improved, but the authors appear to have only been moderately responsive to the Reviewers' prior concerns and requests.

I can only apologise for appearing to be only moderately responsive. I can assure you that we made every effort to genuinely address specific concerns detailed after the first review.

I agree with Reviewer 1's concerns about creating a ranking or at least a 2-4 tier ranking of the identified biomarkers to better enable readers to benefit from and potentially make use of the detailed analyses done by the authors.

I also agree with Reviewer 1's request to present the biomarkers by BOTH this ranking AND separated by their source material, as the potential future utility (and commercial viability) of the identified biomarkers may be different depending on their source.

We have taken these comments on board and re designed the layout of the table. Biomarkers are now separated into different tables according to whether they are predictive or evaluative and their source (Figure 5). We thank you for the suggestion. 1

This manuscript describes a nice approach to evaluate biomarkers to distinguish SRNS and SSNS. It could have benefitted however from the inclusion of a clinical nephrologist as a co-author both to improve the readability and clinical relevance of the identified findings.

Professor Gavin Welsh (professor of renal cell biology) and Professor Moin Saleem (professor of renal medicine and consultant paediatric nephrologist) have been added as authors and have contributed to the manuscript. Both are world renowned experts in podocyte biology and the pathogenesis of idiopathic nephrotic syndrome. They have suggested edits throughout the manuscript which will be evident in the tracked changes. 

It also should be submitted double-spaced and include page numbers and line numbers to facilitate review of the manuscript. 

Thank you, this has been done.

1 - The authors need to add a paragraph to the discussion addressing the potential impact of confounding of the results presented in their study due to the presence vs. absence of ongoing immunosuppressive treatment of NS on the reported values for biomarkers of NS, as few of the reported studies controlled for the presence vs. absence of immunosuppressive treatment, or even addressed this concern as a major potential source of confounding of their data.

The following paragraph has been added to the discussion. 

“While some of the presented studies controlled for the exposure of subjects to immunosuppressive treatment, many did not. It is vital when looking for biomarkers of steroid resistance to receive sufficient information concerning treatment to understand the context of the biomarkers in those subjects. Given the paucity of information in many of the identified studies concerning the treatment regimen, the authors recommend that future studies pay particular attention to this aspect of their biomarker studies.”

2 - Please add more details about the NS subgroups described midway through the Introduction, and their relative risks for progression to kidney failure.

I have added the following paragraph: “NS can also be characterised by its histopathological features. Focal Segmental Glomeruloscelrosis (FSGS) progresses more rapidly to end-stage renal failure compared to minimal change disease (MCD) [17, 18]. Histopathological variants have limited correlation with the pathogenesis of the different NS entities, however, renal biopsies of SRNS generally show FSGS [19].”

3 - Please add the details promised to Reviewer 1 in your rebuttal letter to the Methods section.

I have further elaborated on the extra detail provided pertaining to the BIOCROSS assessment as follows: “Each of the 10 items has three issues to consider. If each issue is covered then the publication will score 2 for that item, if only one or two of the issues are covered then the paper will score 1 and if none of the issues are covered then the score will be 0. A total score of 20 is available for papers that cover all the issues across all items and domains.”

4 - In Figure 1, please add details about the ranges and interpretation of the BIOCROSS scores in the table to improve reader interpretability.

I have marked within the table itself that the BIOCROSS scores are out of 20 and then I have added the following underneath the table: “BIOCROSS marks are out of 20 and scored across 5 domains: study rationale, design/methods, data analysis, data interpretation and biomarker measurement.”

5 - Figure 2 is too small for the readers to see. Please either enlarge it to make it readable or consider omitting.

Thank you for the feedback we have enlarged it. 

6 - Figure 3 (4?) should include the names of each dot directly in the figure, and the legend should precisely clarify the meaning of the QUADRAS scores shown in the figure to improve readability.

We have tried to accommodate your request but have been unable to achieve this in a satisfactory way. Some of the datapoints are clustered to the point that adding in any extra information (such as the full name and description of the biomarker or panel) actually diminishes the readability. We have put a key underneath the full figure with the full names of each biomarker. We feel this is the best compromise. This figure is not showing QUADAS scores, it is showing the specificity and selectivity for the biomarkers for SRNS. 

For figure 3 I have added the following paragraph underneath to help the readability: “For each signalling question we have looked at the appropriate papers and assessed if they have answered the question. Responses that reduce the risk of bias have been marked in green while those that possibly increase bias have been marked in red. These have been taken together to assess the overall risk of bias.”

7 - As also promised in the rebuttal letter, please spell out the full names of all biomarkers followed by the acronyms after their first usage consistently throughout the manuscript.

We have ensured that this has been done. Moreover, we now include a list of abbreviations at the start of the manuscript.

---

## [Editor Report · Decision Letter 2]

4 Oct 2024

Biomarkers to predict or measure steroid resistance in idiopathic nephrotic syndrome: a systematic review.

PONE-D-23-12560R2

Dear Dr. Carl James May

We’re pleased to inform you that your manuscript has been judged scientifically suitable for publication and will be formally accepted for publication once it meets all outstanding technical requirements.

Kind regards,

Rajendra Bhimma, PhD

Academic Editor

PLOS ONE

Additional Editor Comments (optional):

Thank you very much for addressing the concerns of the reviewers.
---

## [Editor Report · Acceptance letter]

18 Oct 2024

PONE-D-23-12560R2 

PLOS ONE

Dear Dr. May, 

I'm pleased to inform you that your manuscript has been deemed suitable for publication in PLOS ONE. Congratulations! Your manuscript is now being handed over to our production team.

Kind regards, 

on behalf of

Professor Rajendra Bhimma 

Academic Editor

PLOS ONE